# Vision-Based Eye Image Classification for Ophthalmic Measurement Systems

**DOI:** 10.3390/s23010386

**Published:** 2022-12-29

**Authors:** Giovanni Gibertoni, Guido Borghi, Luigi Rovati

**Affiliations:** 1Department of Engineering “Enzo Ferrari”, University of Modena and Reggio Emilia, 41125 Modena, Italy; 2Department of Computer Science and Engineering, University of Bologna, 40126 Bologna, Italy

**Keywords:** pupillary light reflex, ophthalmic instrumentation, eye status classification, computer vision-based classification, machine learning, deep learning, expert systems

## Abstract

The accuracy and the overall performances of ophthalmic instrumentation, where specific analysis of eye images is involved, can be negatively influenced by invalid or incorrect frames acquired during everyday measurements of unaware or non-collaborative human patients and non-technical operators. Therefore, in this paper, we investigate and compare the adoption of several vision-based classification algorithms belonging to different fields, i.e., Machine Learning, Deep Learning, and Expert Systems, in order to improve the performance of an ophthalmic instrument designed for the *Pupillary Light Reflex* measurement. To test the implemented solutions, we collected and publicly released *PopEYE* as one of the first datasets consisting of 15 k eye images belonging to 22 different subjects acquired through the aforementioned specialized ophthalmic device. Finally, we discuss the experimental results in terms of classification accuracy of the eye status, as well as computational load analysis, since the proposed solution is designed to be implemented in embedded boards, which have limited hardware resources in computational power and memory size.

## 1. Introduction

Nowadays, a popular research field is the integration of vision-based solutions into the instrumentation and measuring industry [1]. Indeed, *Artificial Intelligence* (AI) applied to *Computer Vision* [2], in terms of *Machine Learning* (ML) [3], *Deep Learning* (DL) [4], and *Expert Systems* (ES) [5], can be effectively exploited to improve the performance of traditional instrumentation systems [6]. More specifically, the use of AI has become a common practice in biomedical imaging [7,8], as well as in ocular data analysis [9,10] and ophthalmic image analysis [11,12,13]. Indeed, the use of image acquisition devices, such as digital cameras, in ophthalmic imaging instrumentation has introduced several improvements, especially for the diagnostic and objective detection of the progression of a number of pathologies [14]. The aforementioned ophthalmic procedures and systems comprise, among others, slit lamp microscopy [15], optical coherence tomography (OCT) [16], funduscopy [17], pupillometry [18], as well as laser eye treatment for the correction of myopia, hyperopia, glaucoma [19], or cataract surgery [20].

In our work, we consider the image acquisition system of an ophthalmic instrument (an in-depth description is available in the original paper [21]) specifically designed and realized for the analysis of *Pupillary Light Reflex* (PLR) [22], i.e., the measurement of the diameter of the pupil in response to specific light stimulation. A schematic drawing of the instrument is depicted in Figure 1. As shown, its working principle, further detailed later in Section 3.2, is based on the acquisition of eye images during light stimulation of the retina, and therefore, its accuracy is often compromised by invalid or inaccurate recordings performed during laboratory measurements due to unaware or non-collaborative human patients or the incorrect use of the instrumentation by non-technical operators. Therefore, we aim to exploit vision-based classification techniques in order to discard non-correct acquired images and then help ophthalmologists or researchers in acquiring more consistent data traces useful for future investigation and research activities.

Specifically, in this paper, we investigate and compare several vision-based algorithms belonging to different research fields, i.e., ES, ML, and DL, for the classification of eye images, especially focusing on the trade-off between classification accuracy and computational complexity. Indeed, in addition to the final accuracy, speed performance is a key aspect of the investigated solutions since, nowadays, advanced ophthalmic exams and procedures need to be performed on compact or embedded systems [23] with real-time performance. Among the requirements, the compactness and portability of the system are important elements that can potentially limit the computational capability of such systems.

In summary, the contributions of the paper are listed in the following:We carry out an extensive experimental evaluation and comparison of different vision-based classification algorithms belonging to different domains, i.e., Expert Systems (ES), Machine Learning (ML), and Deep Learning (DL), to classify whether an input eye image is correctly acquired or not;To test the investigated solutions, we collected and publicly released a new dataset, namely *PopEYE*, consisting of about 15 k images belonging to 22 different subjects (16 males, 6 females). To the best of our knowledge, this is one of the first datasets with this kind of data acquired through a professional ophthalmic system for *Pupillary Light Reflex Measurement* (PLR);We analyze and discuss the obtained results in terms of final accuracy and real-time performance. Experiments related to speed performance are conducted on three different commercially available solutions: a high-end desktop computer, a laptop, and an embedded board (*Raspberry Pi*).

This paper is organized as follows. First, in Section 2, we discuss related works in which a vision-based approach has been adopted together with eye images and, more specifically, for eye-gaze, eye-tracking, and pupil localization. In Section 3, we present a description of the experimental setup and the procedure used for collecting and classifying the eye images. Section 4 illustrates the specific vision-based classification algorithms belonging to ES, ML, and DL used for image classification. The results in terms of validation accuracy and prediction time obtained by exploiting all the approaches previously described are presented in Section 5. The paper concludes, after the presentation of the prospective research activities, in Section 6.

## 2. Related Works

Many studies concerning eye imaging published in the last few decades are mostly focused on vision-based techniques aiming to fulfill complex tasks, such as eye tracking [24], gaze estimation [25], pupil, and iris recognition [26,27]. With this goal in mind, numerous researchers worked toward the collection of a large number of datasets and in some cases, helped make those datasets available to the public. Unfortunately, it should be noted that for each specific task and dataset, the eye images can be characterized by a number of different factors, including the viewing angle and framed field, head position, eye color, and light circumstances, as well as eye state (i.e., open, half closed or fully closed), skin texture, ethnicity and age of involved subjects, and the visual state of health.

According to the research conducted by Winkler et al. [28], the bulk of eye-tracking datasets gathered between 2004 and 2012 had an average viewing distance of 70 cm, with frames recorded in a laboratory environment using a chin rest, a biting bar, or head-mounted devices to limit the patients head movement. Other datasets, instead, include face and eyes images collected in natural environments [29] or with commercially available devices, such as smartphones [30], thus to solve most common eye gaze estimation during everyday tasks, such as driving [31], reading [32], or browsing the internet [33]. Other studies include images for the detection of natural blinks [34,35], some of them with the aim of improving gaze estimation [36].

Differently, datasets used for pupil localization and iris recognition only partially frame the subject’s faces, as most of them exploit custom-made head-mounted devices to acquire eye images from a closer viewing angle [37,38]. Recently, the work published by Omelina et al. [39] reviewed tens of datasets concerning iris recognition tasks. Although many of these datasets are now freely available online, none of them were appropriate for our specific purpose due to the lack of mandatory annotations or labels, as well as the specificity of the images we required in our test case that aim at improving the acquisition quality of a *Pupillary Light Reflex* instrument.

Several works that tackled the task of pupil detection, eye localization, or face recognition have been introduced and improved over the years. Timm et al. [40] suggested a method to localize the eye center by means of gradients. Other approaches aimed to further investigate this task. An algorithm based on a k-means histogram segmentation proposed by Swirski was able to achieve 50 fps on a desktop CPU with a 5 px error on highly off-axis images [41]. Others [42] tried to improve the efficiency of the localization problem on the BioID face database [43] with a combination of Haar features and efficient SVM, thus reaching 10 fps and 90% accuracy (e≤0.1) on a 3.0 GHz desktop CPU. Lately, the research group of Kacete proposed a Hough randomized regression tree obtaining an overall accuracy of 97.9% with e≤0.10 at around 30 fps on a desktop CPU.

Following that, numerous different forms of Deep Learning algorithms, notably Convolutional Neural Networks (CNN) [24,44,45,46], were proposed by researchers to obtain greater localization accuracy over more heterogeneous datasets, all performing at around 90% depending on normalized error threshold *e*, at the cost of increased computational complexity. Indeed, the best results were reached with *DeepEye* [45], with an 87% detection rate with 5 px error while performing around 32 fps with the employment of an Nvidia Tesla K40 high-end GPU. More recently, the combination of Deep Learning and the Expert System algorithm presented by Khan [47] showed 100% accuracy with e<0.1 on popular datasets Talking-Face [48], BioID [43], and GI4E [49], thus without reporting any real-time metrics. In Table 1, we summarize the most effective vision-based face and eye recognition techniques presented in the last 10 years.

Besides these, different techniques, such as pseudo-depth mapping from RGB images [51] or the implementation of a two-stage training process with anomalous data samples [52], have been investigated to further increase the image recognition accuracy and generalization capabilities of commonly used CNN architectures. While many studies have been carried out to improve face and eye detection, or pupil tracking with vision-based techniques, it is worth noting that most investigations focused on the achievement of lower localization error, i.e., higher detection rates and accuracy, at the cost of higher computational complexity. Moreover, our findings show that there is a lack of performance comparison for the majority of the aforementioned approaches on different platforms. In our opinion, this is a key aspect that can assess the scalability of such systems for more simple, effective, and efficient solutions.

## 3. PopEYE Dataset

In this section, a short description of the instrument setup and the acquisition procedure is presented. In particular, in Section 3.1, the acquisition device used for the experiment is briefly described. Moreover, the test methodology adopted and the technical procedure to acquire and manually annotate eye images are illustrated in Section 3.2.

### 3.1. Acquisition Setup

The system used for this experiment is a *Maxwellian-view*-based monocular pupillometer [53] with a configurable BGRY LED-based light source and a board-level CMOS camera used to acquire grayscale images of the human eye. As described in [21], the optical design of the instrument presents several advantages in the field of pupillometry. In particular, the two lense stages allows precise control of the light stimulation level of the retina while avoiding the utilization of pupil dilation. The device schematic diagram is depicted in Figure 1; it consists of a double-lens optical system, including four fiber-coupled high-brightness LED sources. Lens **L1** ($f_{1}=40$ mm, $D_{1}=2.54$ cm), together with a stop aperture, has been used to collimate the light towards lens **L2** ($f_{2}=50$ mm, $D_{2}=2.54$ cm) that focuses the light beam to the eye’s pupil plane, obtaining an output spot-size of ≈900 μm diameter with a full width viewing angle of $\theta =20.5^{\circ }$. The proper functioning of the system, i.e., correct visualization of the stimulus by the subject, requires the eye to be precisely aligned to the center of lens **L2**. Moreover, since the PLR is observed through lens **L2**, an incorrect alignment of the eye, and therefore, a shift between the center of the pupil with respect to the optical axis of the lens would introduce spherical aberrations in the recorded image, which would modify the shape and size of the observed pupil, resulting in incorrect measurements.

The eye images are acquired thanks to the illumination performed by eight 850 nm infrared (IR) LEDs circularly arranged around lens **L2**, while a dichroic mirror (DMSP750B, Thorlabs ^©^ Newton, NJ, USA) has been used to reflect the pupil image at $90^{\circ }$ toward the compact high-resolution camera (mvBlueFOX3-3M, Matrix Vision GmbH Oppenweiler, Germany GE) equipped with a 10.1 mm f/2.8 lens (Edmund Optics Inc. Barrington, NJ, USA). This allows the separation of the *acquisition path* from the *stimulation path*, which is necessary for recording in-axis images of the eye. An infrared long-pass filter ($\lambda_{cutoff}=750$ nm) has been additionally used and positioned in front of the camera lens to filter out the visible light, making it possible to have fixed illumination of every frame, even during different types of ongoing light stimulation.

A desktop computer running a custom-developed program allows for capturing images of up to 60 fps. The camera is hardware triggered by the micro-controller unit to synchronously acquire frames with light stimulation coming from the high-brightness LEDs. An interface designed with NI LabVIEW allows the user to control and configure the type of stimulation and monitor in real-time the pupil response dynamics.

### 3.2. Acquisition Procedure

The pictures used for the investigation have been collected from both the left and right eyes of 22 subjects, 6 female and 16 male, with ages between 20 and 38 years old. The pupillometer is set to acquire continuous monochromatic 8-bit depth pictures at 50 Hz. The output frames recorded from the sensor (SONY IMX178 6.4 Mpx CMOS) have been resized and stored as uncompressed video files with an output resolution equal to 772×520 pixels (width and height, respectively).

During the acquisition, we asked the subjects to observe a circular light target of fixed light intensity and under voice command, to move the gaze in the four cardinal directions: right, down, left, and up. Moreover, we recorded the subject’s natural eye blinks as well as a slowly controlled one, i.e., while asking them to close their eye completely and slowly open their lids. Thereafter, individual frames were extracted from the saved video files with a temporal sub-sampling with a sample rate of 10 Hz, thus avoiding nearly equal frames from a visual point of view. Each frame has been manually labeled into one of six different classes: “correct”, “closed”, “right”, “down”, “left”, and “up”. Sample images that illustrate each class are shown in Figure 2.

As shown, every image contains the eyeball as seen through lens **L2** (see Figure 1). The region outside the lens area is dark as it is not illuminated by the infrared LEDs. Manual classification has been carried out by the same operator, with the help of visual aids depicted on the acquired images. In particular, as shown in Figure 3, each frame has been cropped to a square-shaped image (520×520 pixels) and divided into four zones obtained by the two diagonals. Moreover, a circular area (with a radius set to 69 pixels) has been added to define the *central region* to support the annotation phase. The size of the *central region* has been chosen specifically to avoid spherical aberrations of the pupil and non-uniform illumination of the retina by means of the light beam used for the stimulation. The ES algorithm, described in Section 4.1, is also used during the annotation procedure to highlight, if possible, the position of the pupil (denoted with a blue bounding box in Figure 3) and its center (yellow point) to support the final labeling operation.

The operator, supported by the aforementioned fixed visual aids, evaluates the location of the center of the pupil and assigns one of the six labels as follows:**Correct**: the center of the pupil is inside the central region (edge included), and the pupil area is uncovered (>90% is visible);**Closed**: the center of the pupil is inside the central region (edge included), and the pupil area is covered by the eyelids (>10% covered), the pupil center is outside the central region and covered by the eyelids (>85% covered), or the pupil is not visible or detectable by the operator;**Up, right, down, or left**: the center of the pupil is outside the central region, and the pupil area is visible or partially covered by the eyelids (>15% is visible); the final class depends on which triangle the center of the pupil falls into.

For each frame, the ES algorithm automatically suggests the position of the pupil and then the label that should be assigned following the previous rules. The operator must manually confirm or change the proposed label, and in the eventuality that the ES algorithm cannot find the pupil or the detection is not accurate, the human operator must choose a label on their own based on the aforementioned rules. The obtained dataset, named *PopEYE*, constitutes a total of 14,976 images, and has been divided into classes as indicated in Table 2. The unbalanced data distribution represents a challenging element since it can negatively affect the training phase, especially for Machine and Deep Learning approaches [54].

## 4. Vision-Based Automatic Eye Image Selection

In this section, we describe the vision-based algorithms used to tackle the image classification problem described in Section 3. Among others, we investigate methods belonging to Expert Systems (ES), Machine Learning (ML), and Deep Learning (DL). Figure 4 visually summarize the investigation conducted in the paper: after the image acquisition, the three aforementioned approaches have been tested, in order to obtain results in terms of classification accuracy, Single Frame Prediction Time (SFPT), and a specific combination of the two with the Figure of Merit (FOM) introduced for this study.

### 4.1. Expert Systems

The large availability of open-source packages, such as the *Open Source Computer Vision Library* (OpenCV) or the *Python Imaging Library* (PIL), offers many functions that simplify the work of solving a variety of Computer Vision and Image Processing tasks.

In our case, we aim to code an algorithm based on expert and a priori knowledge to make the final classification, then use an approach commonly referred to as “Expert System”. Specifically, the proposed method is able to automatically detect, through several sequential image processing operations, the eye’s pupil and its position respective to the center of lens **L2**, i.e., the optical axis of the system (see Figure 1). In the following, we list and analyze the steps of the proposed algorithm, also graphically summarized in Figure 5, to improve understanding and clarity.

The original grayscale 772×520-pixel frames is cropped to a square image of 520×520 pixels, in which the large part of the black borders is removed. The crop coordinates are fixed since the black borders on the image are produced by the acquisition device and then are always in the same position.In order to mask the **L2** lens profile, as visible in the second image of Figure 5, a fixed threshold value set to 80 is applied. Then, if a pixel value is smaller than the threshold, it is set to 0. Otherwise, it is set to a maximum value (255). Finally, the resulting image is normalized by stretching the minimum and maximum values to 0 and 255, respectively.A filter with a *Gaussian* kernel with a size of 21×9 is applied to the image. The greater horizontal dimension for the kernel has been found empirically, and it is specifically used to enhance the blur on eyelashes, which can possibly cover the dark pupil.A second fixed threshold has been used to further remove unwanted brighter parts of the image. The threshold is chosen between the value 70 and the 5-th percentile of the gray level image. Then, the normalization applied in the second step is used a second time, stretching the minimum and maximum values to 0 and 255, respectively.In the resulting image, an adaptive threshold operation is applied; the neighborhood area from which selects the threshold value as 9×9 pixels;The outer ring of the detected lens edge is removed by applying a threshold of pixels located at a certain distance from the image center. Further, in this case, the lens edges are in the same position, and then a fixed distance can be set.The morphological opening and closing operations with an elliptical kernel with size 21×21 have been used to group small dark regions (at this point, the main dark area is the pupil, if present, and eyelash) of the image;Lastly, we find the contours in the binary image highlighted with blue dots in Figure 5. The contours found are subjected to a hull transformation and analyzed in terms of roundness, average brightness, and size. Specifically, the radius of the circumference inscribed in the bounding box computer for each contour must be in the range [20,150], and the average pixel value must be greater than 50. In this way, too-small and too-big contours and to-bright contours are discarded by the algorithm. Then, a numeric score is associated with the remaining contour, i.e., the most circular, darkest, and biggest contour is chosen as the best candidate. Finally, its center is used to determine the pupil’s position.

It is worth underlying that all the previously hard-coded functions, including kernel sizes, threshold levels, and brightness or contour sizes, have been empirically set for our specific test dataset. Any changes in the image’s sizes, aspect ratio, exposure, or contrast will require a manual re-tuning of all these parameters.

According to the rules outlined in Section 3.2, by evaluating the position and the aspect ratio of the pupil found by the algorithm, the predicted label is assigned to each frame automatically. If any pupil contours are found in the last step, the frame is labeled as *closed*.

### 4.2. Machine Learning

Within the *Machine Learning* field, we test a variety of classifiers in combination with two different hand-crafted features applied to the images, i.e., *Histogram of Oriented Gradient* [55] (HOG) and *Local Binary Patterns* [56] (LBP).

Specifically, HOG features have been originally created for the task of *People Detection* [57], but they have been successfully used in a variety of different tasks [58] in the computer vision community. HOG encodes the information of the edges’ orientations into an image, and then it can be useful in our classification task since the visual appearance of the eye is strictly related to the position of the edges. In our experiments, we produce the final HOG histogram with eight orientations, computed on 4×4 pixels per cell and 1×1 cells per block.

Moreover, LBP features have been widely used in the literature and in the Computer Vision community, and, in particular, to have a powerful descriptor of microstructures (texture) present inside an image. For this reason, we believe that this hand-crafted feature can be effectively used in our classification task about the eye status, even though it has a spatial support area and the noise in the image can affect two single-pixel values [59]. In our experimental procedure, we empirically code the LBP feature, setting the radius R=5 and the number of sample points P=100.

In our experimental investigation, we specifically focus on eight different classifiers, namely *Support Vector Machine* [60] (SVM), *Multi-Layer Perceptron* (MLP) [61], *Random Forest* [62] (RF), *Decision Tree* [63], *Gaussian Naive Bayes* [64], *K-nearest neighbors* [65], *Adaptive Boosting* [66], and *Quadratic Discriminant Analysis* (QDA) [67]. With these classifiers, we aim to investigate not only the final accuracy but also the computational load.

We use the SVM classifier with a *Radial Basis Function* (RBF) [68] kernel, the regularization parameter C=1.0, and the kernel coefficient $\gamma =10^{-3}$. For the MLP classifier, we adopt an architecture composed of three layers (1 input, 1 output, and 1 hidden layer with 100 neurons), *ReLU* [69] activation, and the *Adam* [70] optimizer. With the RF ensemble classifier, we set the number of trees in the forest equal to 100, using *Gini*’s index [71] to measure the quality of the obtained splits (also used with the Decision Tree classifier). For the Gaussian Naive Bayes, we set the smoothing value to $e^{-9}$, while we used the K-Nearest Neighbors with the number of neighbors equal to 3, and the weight function used in the prediction set to *uniform*. All these classifiers are tested with the aforementioned hand-crafted features as input, in addition to the case in which the input image is simply unrolled (for instance, an input image of 32×32 images is flattened in a vector of 1024 elements) and normalized by a factor of 255 (the maximum value of a given pixel), putting the input data in a range of [0,1].

### 4.3. Deep Learning (CNN)

In the Deep Learning approach, we select and compare several different architectures of *Convolutional Neural Networks* (CNNs).

The first is the *MobileNet* [72] architecture since it is a well-known architecture that combines a great overall accuracy on the reference dataset *ImageNet* [73] and a limited number of internal parameters (4.3 M). This element leads to a limited computational load on the GPU on which the model is running and a reduced amount of training data. In our implementation, we set α=1, the delta multiplier to 1, and the dropout to $10^{-3}$ [74].

The second exploited CNN is *ResNet-50* [75], an architecture widely used for different classification and regression tasks, with good accuracy and a greater amount of parameters (25.6M). For similar reasons, we also test the VGG neural network, one of the most well-known object recognition models available in the literature. Specifically, we use the architecture with 16 layers, usually referred to as the VGG-16 model. Differently from *MobileNet*, these architectures can suffer from the limited availability of training data and are more complex from a computational point of view.

Among the other neural networks, we test, i.e., Inception-V3 [76] and EfficientNetB0 [77]. In the first case, we observe that the presence of multiple filters with different size at the same level can contrast the overfitting phenomena [78], especially when a limited amount of training data are available. In the second case, EfficientNet has been proven to offer great accuracy in classification tasks, with an order of magnitude fewer parameters with respect to common CNNs.

In both cases, with respect to the original architecture, we change the last fully connected layer with a dense layer with six neurons, which is equal to the number of our dataset’s classes. To fine-tune both neural networks, pre-trained on *ImageNet* [73], we exploit the *Stochastic Gradient Descent* (SGD) as an optimizer, with a learning rate equal to $10^{-3}$ and a momentum of 0.9. We exploit the Binary Cross-Entropy (BCE) loss function, defined as follows:(1)$$BCE=-\frac{1}{N}\sum_{i=1}^{N}y_{i}\log\left(p\left(y_{i}\right)\right)+\left(1-y_{i}\right)\log\left(1-p\left(y_{i}\right)\right)$$
where $y_{i}$ is the binary label of the *i*-th sample, *N* is the total number of data points, and $p(y_{i})$ is the predicted probability of a certain sample belonging to a specific class. In our experiments, we exploit the *TensorFlow* [79] framework with *Keras* [80] wrapper implementations.

## 5. Results

In this section, we present all the results obtained through the algorithms described in Section 4. All the approaches have been tested on the same test dataset constituted by 30% of the entire dataset, i.e., 750 images for each class. It is important to observe that the test dataset is balanced (simplifying the score evaluation and understanding), while the training dataset, used only with ML and DL (since ES does not require training data), is highly unbalanced and constituted, as reported in Table 3.

Moreover, as introduced before, one of our goals is to evaluate the computational cost of the proposed algorithms on different systems, thus investigating the eventuality of more portable and more efficient solutions. Therefore, we compared the *Single Frame Processing Time* (SFPT) among the systems described in Table 4. The SFPT consists of the sum of the elapsed time between the instant in which the frame is available in the RAM memory and the time instant in which the final classification is produced by the system. We observed that, in our scenario, the acquired frames are used in real-time and then the *load time*, intended as the elapsed time necessary for the system to load the frame from the local hard drive to the RAM, should be neglected. In all approaches described below, the SFPT measurement has been computed as the average on the entire *test dataset*, i.e., 750 frames per class for a total of 4500 images. In addition, we also report the 90th percentile of the SFPT time in order to highlight the impact of higher prediction times due to variable loads that can affect the performance of the tested systems.

### 5.1. Expert System Algorithm

For the ES algorithm described in Section 4.1, the experimental results are reported in Table 5.

As shown, the proposed algorithm is able to achieve over 90% accuracy using full-size input images. Unfortunately, we observe that the proposed algorithm relies on a series of values, thresholds, and hard-coded image processing procedures that strictly depend on the dataset and images acquired. These elements can limit the generalization capability of the developed algorithm and negatively affect the speed performance, as listed in the second and third columns of Table 5.

### 5.2. Machine Learning-Based Approaches

The classification accuracy obtained with the best-performing Machine Learning-based methods in combination with different hand-crafted features is reported in Figure 6.

As shown, from a general point of view, all six classifiers, i.e., MLP, SVM, RF, Decision Tree, GaussianNB, and KN-neighbors, obtain good accuracy in the range of [80%, 100%]. The most interesting element to note here is the use of unrolled images as input, without the computation of any manual feature: in all cases, this (very) simple image feature representation is the best solution in terms of accuracy. As shown, the use of HOG features produces better results with respect to LBP features, widely used to code the texture information in images (while HOG mainly encodes the shape). The LBP feature, in combination with the SVM classifier, achieves the worst accuracy. It is also important to note that the input size is only slightly related to the final accuracy: this is an important finding since, as reported in Figure 7, large input sizes negatively affect the speed performance. Among the different ML classifiers, we also tested the Adaptive Boosting and Quadratic Discriminant Analysis classifiers. However, we have decided not to report the results relative to these two as they showed the worst performances in terms of accuracy [<80%]. In terms of speed performance, for all six classifiers and image resolutions, LBP features represent the worst scenario, followed by the HOG features. As expected, unrolled images as the input have the lightest impact on the computational load since no particular operations are needed: this element, combined with the great accuracy highlighted in the previous analysis, promotes the use of MLP or SVM classifiers with unrolled images as input data as the best solution within the ML scenario.

### 5.3. Deep Learning-Based Approaches

The results obtained through the Deep Learning approach are reported in Table 6. As shown, almost all the tested Neural Networks achieve great accuracy, even with a limited amount of training data: in this context, we observe that the use of pre-trained models is fundamental in order to obtain good accuracy. As was foreseeable, the approach based on deep neural networks achieves the best accuracy across all the other approaches: the ability to automatically learn features from images is a key element in our classification task.

On the other hand, DL offers the worst speed performance, while the use of pre-trained networks forces the use of input images with 224×224-pixel resolution and 299×299 for Inception V3. The best speed performance in terms of SFPT has been obtained with *MobileNet* (only 6 milliseconds of difference in respect to the best performing ResNet50 with the Desktop configuration as reported in Table 4), at the cost of a very minimal drop in terms of classification accuracy. It is interesting to notice how SFPT, for different network architectures, scales accordingly to the number of parameters and image input sizes, while other factors may affect the time performances as well. It is worth stating that SFPTs have been calculated on the prediction of individual frames one at a time, as that is what would happen in a real-case scenario. This approach significantly impacts the workload on the GPU, as the majority of the time is spent transferring individual frames from the RAM memory to the dedicated graphics memory on the GPU.

### 5.4. Final Considerations

Among all the classification algorithms tested, we have reported the best combination of classification accuracy and speed performance obtained through each proposed approach in Table 7.

More precisely, to compare and select the best algorithms, we used Equation (Equation 2) as a mathematical *Figure of Merit* (FOM):(2)$$FOM=A^{2}\cdot e^{-}{\left(\frac{SFPT}{\tau }\right)}^{2},\;\tau =\frac{-10\times 10^{-3}}{\sqrt{ln\left(\frac{0.5}{0.8^{2}}\right)}}≅0.02013\;s^{-1}$$

The equation gives the score value FOM∈[0,1], where A∈[0,1] is the accuracy, SFPT is the *Single Frame Prediction Time* in seconds, and τ is a constant used to balance the equation in order to get FOM=0.5 for a 0.8 accuracy and SFPT=10 ms, which we consider as the target minimum score. Indeed, target SFPT has been set to guarantee an output frame rate higher than 50 fps to match the standard of modern pupillometer devices [81,82], thus taking into account the additional processing time that can be needed to extract the pupil diameter value from the *correct* frames. For this reason, we designed the equation to penalize the processing time with an exponential decay, as the algorithm is intended to be used in *real time*. As a consequence, a score of FOM<0.5 would be obtained even with A=1 if the SFPT is not small enough ($SFPT_{limit}≃16.75$ ms). Moreover, the accuracy *target* score is set to 0.8 to limit, in the first place, the amount of wrongly discarded images to ($\frac{1}{5}$ or 20%), and secondly, to match the minimum average score of algorithms with a similar task [32,44,83].In the equation, *A* is squared, such as any solutions, even with the lowest SFPT, will give a FOM<0.5 if the algorithm accuracy is lower than ($A_{limit}=\sqrt{0.5}$). Figure 8 shows how the *FOM* from Equation (Equation 2) varies for different values of *A* and *SFPT*.

From the figure, it is clear why the DL algorithm shows the lowest score since while having the highest accuracy (A=99.8%), it is also the most computationally expensive one, making it not adequate for practical implementations. The results reported in Table 7 suggest the ML approach with a SVM classifier and a 32×32 pixels input size image is the one showing the highest score, with an average SFPT of 0.202 milliseconds and an accuracy equal to 96.2% on the desktop computer.

Finally, to further investigate whether these approaches can be employed on less power-consuming and more portable systems, we compute the SFPT on the different systems, as detailed in Table 4. Figure 9 shows the average SFPT comparison among the three devices.

As expected, the SFPT average times for all the methods considered are higher for both the laptop and the *Raspberry* board when compared to the desktop computer. It is interesting to note that ML represents the best solution for speed performance, obtaining real-time on all three tested platforms, and in particular, on the embedded board.

## 6. Conclusions

In this paper, we have investigated the use of vision-based classification algorithms to improve the acquisition quality of a device used for the analysis of *Pupillary Light Reflex* based on the acquisition of eye images. In particular, we have analyzed three different fields, i.e., Expert Systems, Machine Learning, and Deep Learning, aiming to train a classifier able to determine the status of the eye in input images. A new dataset, namely *PopEYE*, is presented and publicly released to test the investigated methods and to obtain the final results. Experimental results, in terms of classification accuracy and speed performance, reveal that the use of Machine Learning-based classifiers with unrolled images as input represents a simple, efficient, and suitable solution, capable of achieving real-time performance, even on the embedded board, *Raspberry Pi*, with limited computational power.

## Figures and Tables

**Figure 1 sensors-23-00386-f001:**
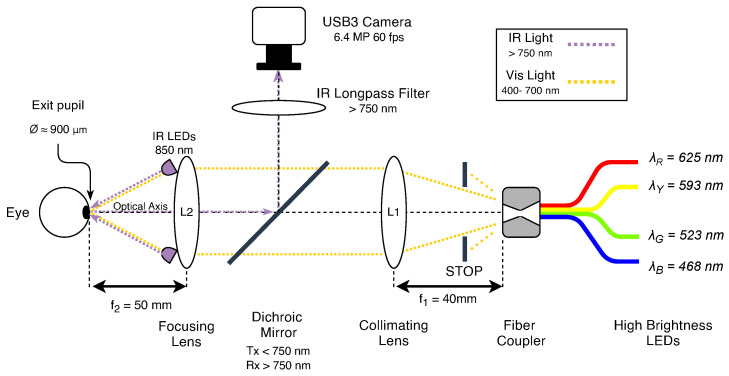
Optical diagram of the *Pupillary Light Reflex* instrument [21] used to collect data. The *Infra-Red Light* optical path (dashed purple line), i.e., the *acquisition path*, allows eye recording while eye stimulation is achieved through the *Visible Light* optical path (dashed yellow line).

**Figure 2 sensors-23-00386-f002:**
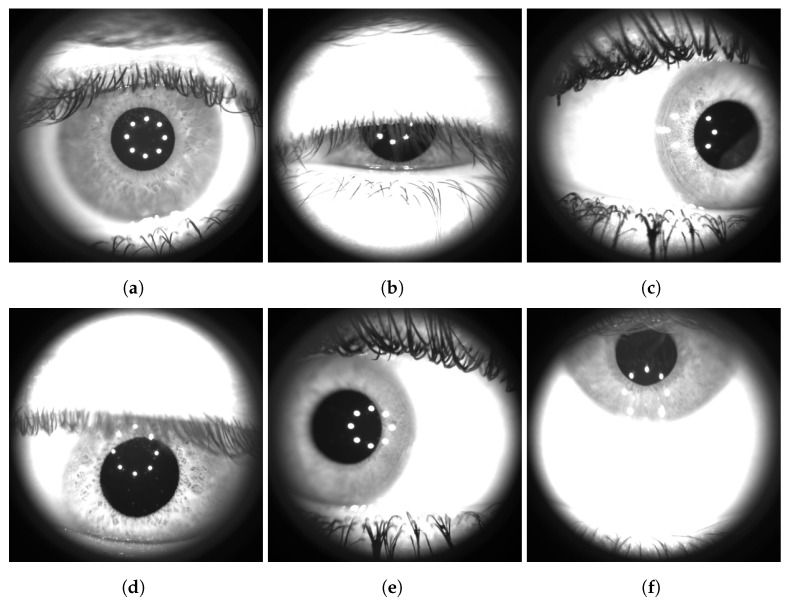
Example of images collected from different subjects of the *PopEYE* dataset. The images have been labeled in six different categories: (**a**) correct, (**b**) closed, (**c**) right, (**d**) down, (**e**) left, and (**f**) up. As shown in the images, the pupil, the iris, the eyelashes and, in certain cases, the eyelid is visible. Visible white spots disposed in a circle near the pupil are reflections produced by the 8 IR LEDs used to illuminate the eye.

**Figure 3 sensors-23-00386-f003:**
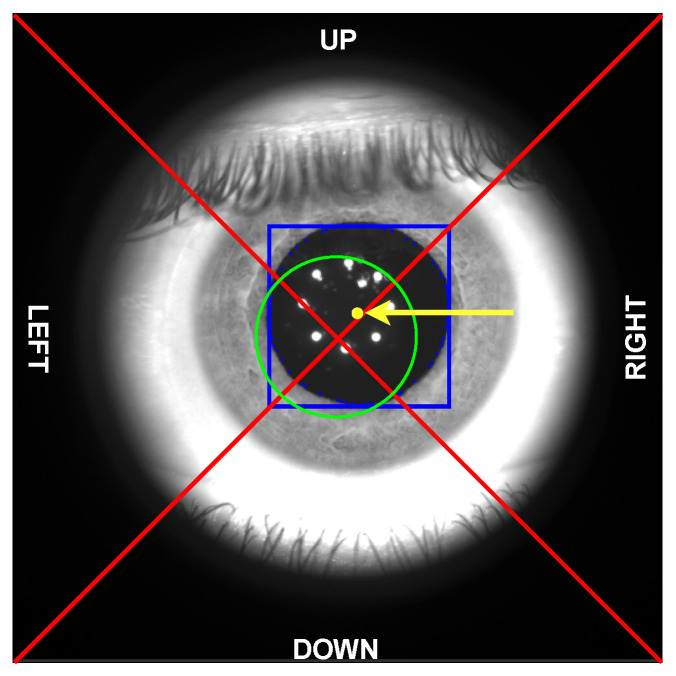
Visual aids used for the data annotation phase. The two diagonals in red define the four cardinal regions, named: *up, right, down, and left*. The green circle determines the *central region* for *“correct”* images. The ES algorithm (described in Section 4.1) finds and highlights the pupil bounding box (blue square) and the pupil center (yellow dot, highlighted by yellow arrow) whenever possible to support the human operator in the labeling process.

**Figure 4 sensors-23-00386-f004:**
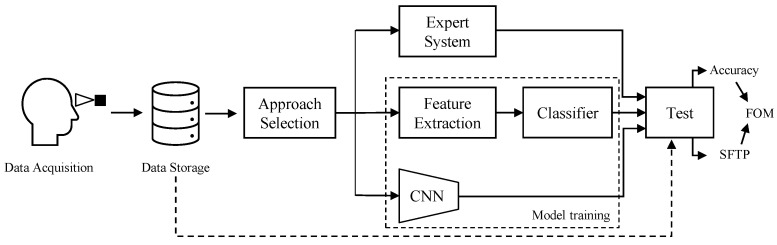
General overview of the proposed analysis. Data are acquired and stored using the device described in Section 3.1 and illustrated in Figure 1. Then, three different approaches based on Expert Systems and Machine and Deep Learning fields are trained and tested on that data, obtaining a final accuracy and SFPT with which it is possible to compute the FOM metric (see Section 5.4).

**Figure 5 sensors-23-00386-f005:**
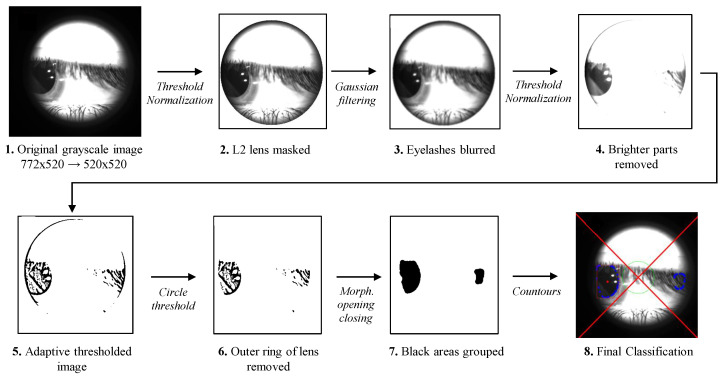
Flowchart of the image processing steps performed by the developed ES algorithm as detailed in Section 4.1. The final detection of the pupil is reported in the last image in which the *pupil center* (red dot) position, in respect to red diagonals and the green circle (see Figure 3) and the aspect ratio of the bounding box (red bounding box) are evaluated; the picture is labeled with the class “left” by the algorithm.

**Figure 6 sensors-23-00386-f006:**
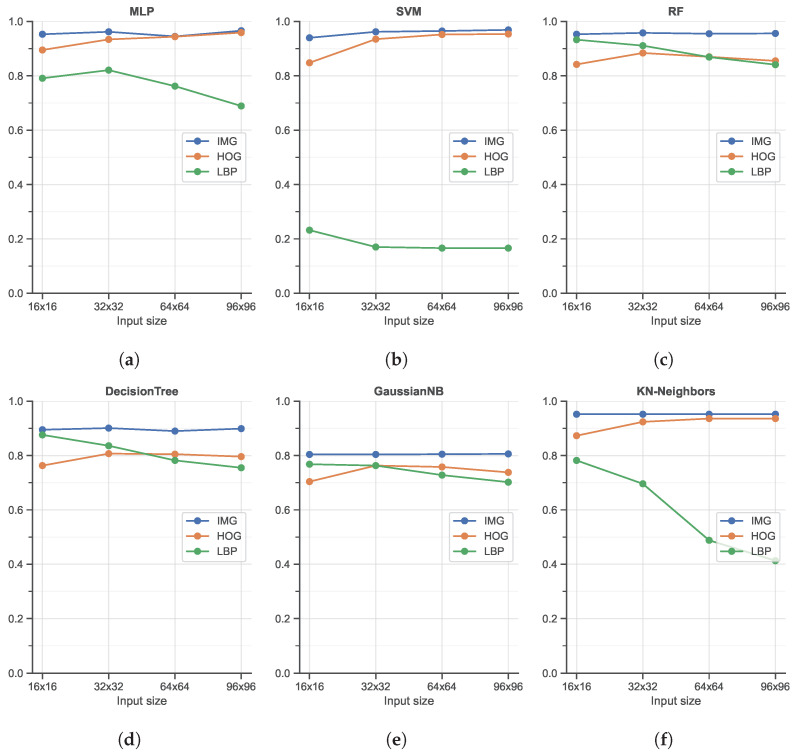
Classification accuracy of the three Machine Learning-based classifiers across different input sizes and input features: (**a**) shows the performance of the Multi-Layer Perceptron (MLP) classifier, while (**b**) shows the Support Vector Machine (SVM) classifier, (**c**) Random Forest (RF), (**d**) Decision Tree classifier, (**e**) Gaussian Naive Bayes (GaussianNB) classifier, and (f) K-nearest neighbors (KN-neighbors) classifier.

**Figure 7 sensors-23-00386-f007:**
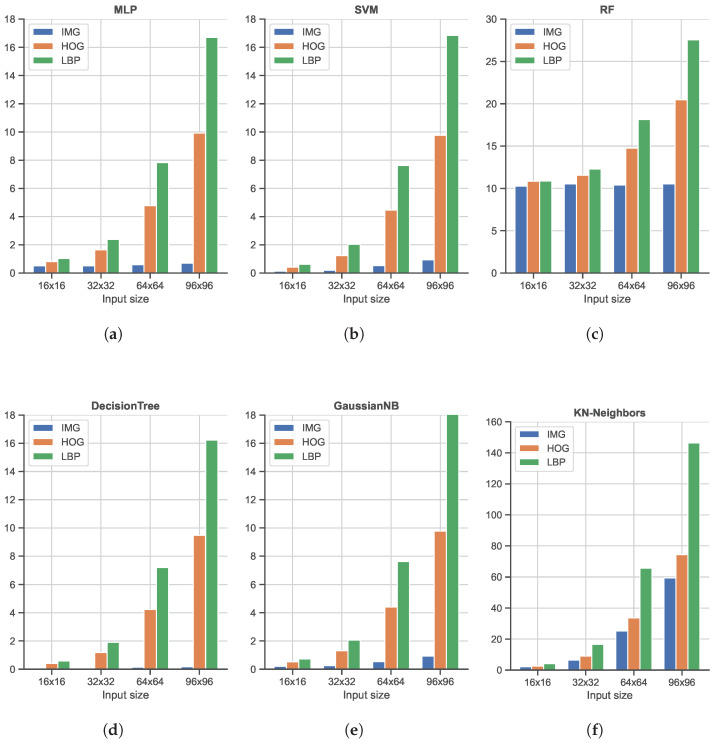
Comparison of the average *Single Frame Processing Time* (SFPT), computed as time consumed for the feature extraction process plus the prediction time, of the three Machine Learning-based classifiers running on a Desktop computer. (**a**) Shows Multi-Later Perceptron (MLP) classifier times, (**b**) Support Vector Machine (SVM) classifier, (**c**) Random Forest (RF), (**d**) Decision Tree classifier, (**e**) Gaussian Naive Bayes (GaussianNB) classifiers, and (**f**) K-nearest neighbors (KN-neighbors) classifier. The time values are expressed in milliseconds (ms).

**Figure 8 sensors-23-00386-f008:**
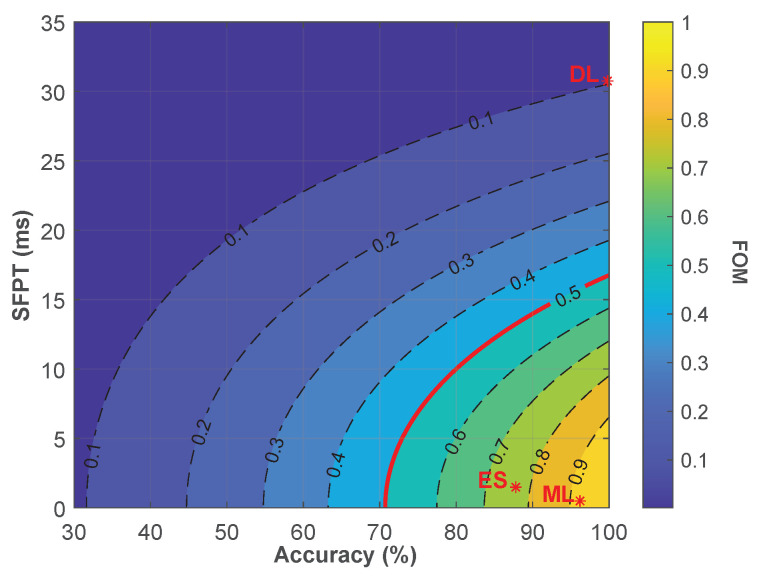
FOM heatmap for different values of A and SFPT. Red stars ***** indicate the score of selected approaches from each category.

**Figure 9 sensors-23-00386-f009:**
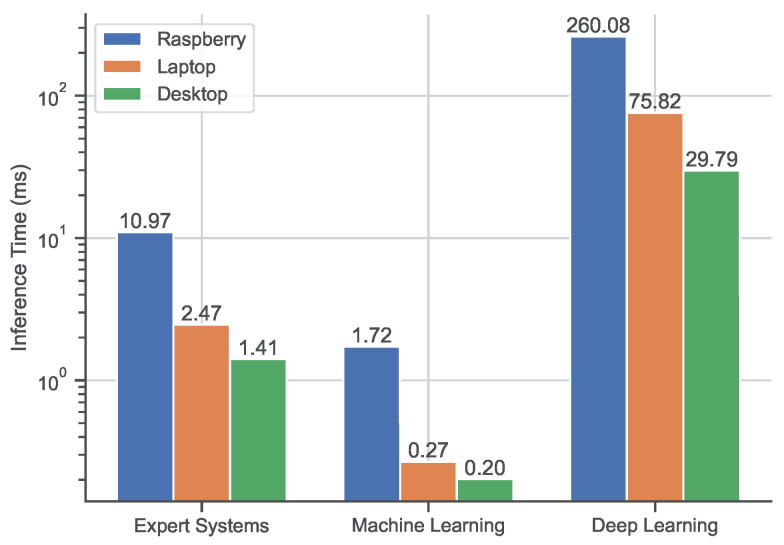
Timing performance comparison across the different platforms. Average SFTP for best solutions of ES, ML, and DL methods. Time values are expressed in milliseconds (ms).

**Table 1 sensors-23-00386-t001:** Summarytable of related methods available in the literature for similar tasks of eye status classification. See Section 2 for further details.

Task	Data	Methodology	FPS	Accuracy (Error)	Architecture	Ref.
Eyes Center Localization	BioID [43]	Gradients	-	93.4 % e≤0.1	-	[40]
Pupil Edge Localization	IR Eye Images	Haar feature, k-means segmentation	50	87 % e≤5 px	Quad-Core 2.8 GHz	[41]
Eyes Center Localization	BioID	Haar Features and eSVM	10	90 % e≤0.1	Pentium 4 3.0 GHz	[42]
Eyes Center Localization	BioID	Hough randomized regression tree	30	98 % e≤0.1	Intel i7 2.7 GHz	[50]
Gaze Tracking	GazeCapture	Deep CNN	10–15	error ∼ 1 cm	ARM1.85 GHz	[24]
Pupil Localization	IR Eye Images	CNN with ASPP blocks	32	87 % e≤5 px	Nvidia Tesla K40	[45]
Pupil Localization	BioID and GI4E [49]	ES and DL	-	100 % e≤0.1	-	[47]

**Table 2 sensors-23-00386-t002:** *PopEYE* dataset composition across the six different annotated classes.

Class	N° Images
Correct	8160
Closed	1790
Right	1336
Left	1296
Up	1379
Down	1015
Total	14,976

**Table 3 sensors-23-00386-t003:** Train and test splitting used in *PopEYE* dataset for both ML and DL approaches.

Class	Train Images	Test Images
Correct	7410	750
Closed	1040	750
Right	586	750
Left	546	750
Up	629	750
Down	265	750
Total	10,476	4500

**Table 4 sensors-23-00386-t004:** Hardware specification of the devices used to compute the SFPT across all experiments.

	Processor				
Device	Name	N° Cores	Max Frequency	Ram	GPU
Desktop	Intel i9-10850K	10	4.8 GHz	32 GB DDR4	NVIDIA RTX 3090
Laptop	Intel i7-7700HQ	4	3.6 GHz	32 GB DDR4	-
Raspberry 4	Cortex-A72 (ARM v8)	4	1.5 GHz	4 GB LPDDR4	-

**Table 5 sensors-23-00386-t005:** Accuracy and timing performance of the ES algorithm on the Desktop Computer.

Input Size	Accuracy	SFPT (ms)	90th Percentile SFPT (ms)
520 × 520	90.2%	10.472	11.904
260 × 260	89.9%	2.160	2.480
130 × 130	87.8%	1.410	1.488
65 × 65	82.1%	1.564	1.984

**Table 6 sensors-23-00386-t006:** Accuracy and speed performance obtained with Neural Network classifiers. Experimental results can also be considered in relation to the size and number of parameters of each neural network.

Architecture	Size (MB)	Parameters	Input Size	Accuracy	SFPT (ms)	90th Percentile SFPT (ms)
MobileNet	16	4.3 M	224×224	99.82 %	29.790	30.752
ResNet50	98	25.6 M	224×224	99.88 %	35.978	36.704
VGG-16	528	138.4 M	224×224	98.80 %	31.473	32.240
InceptionV3	92	23.9 M	299×299	98.14 %	42.149	43.133
EfficientNet	29	5.3 M	224×224	71.81%	38.447	38.689

**Table 7 sensors-23-00386-t007:** Classification accuracy and speed performance comparison across the three investigated approaches.

Approach	Input Size	Classifier	Accuracy	90th Percentile SFPT (ms)—CPU	90th Percentile SFTP (ms)—GPU	FOM
ES	130 × 130	pre-defined	87.8%	1.488	-	0.767
ML	32 × 32	SVM	96.2%	0.496	-	0.925
DL	224 × 224	MobileNet	99.8%	38.688	30.752	0.096

## Data Availability

The full image dataset that supports the findings of this study concerning both the training and validation phases (Section 3) are available from the corresponding author upon reasonable request by email: giovanni.gibertoni@unimore.it.

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
