# Peer review of "Vision-Based Eye Image Classification for Ophthalmic Measurement Systems"

_sensors, 2022, doi:10.3390/s23010386_

Round 1

Reviewer 1 Report

In this paper, the adoption of several vision-based classification algorithms belonging to different fields, i.e., Machine Learning, Deep Learning, and Expert Systems, in order to improve the performance of an ophthalmic instrument designed for the Pupillary Light Reflex measurement is presented. To test the implemented solutions, a publicly released PopEYE dataset is used, as one of the first consisting of 15k eye images belonging to different subjects acquired with specialized ophthalmic equipment.

However, there are a few Major observations based on which this paper cannot be accepted in its current form until it is improved. The comments are as follows.

1.      The introduction and related work sections can be improved by adding more material

2.      The classification diagram of the old and current sub-domains, famous techniques, and algorithms is missing.

3.      Table-explaining the advantages and disadvantages of the existing should be mentioned techniques mentioned.

4.      The author may also categorize different techniques based on performance results and also identify the scenarios where these techniques can be applied. 

5.      The proposed algorithm should be included in the form of a pseudo-code or flowchart

6.      The complete proposed framework is missing both in the form of a detailed diagram and equations.

7.      An experimental setup diagram may also be included which can explain the evaluation model.

8.      Computation time and overhead analysis are missing against the achieved accuracy

9.      Comparison with the similar existing techniques is missing

Reviewer 2 Report

The following comments must be carefully revised to improve the quality of the paper.

1. In Figure 1, the author gives a detailed setup which is impressive. The reasons for adopting such a setup should be further explained.

2. The technical details of the fourth part and the reasons for setting up some hyper-parameters should be added, such as the Gaussian kernel with size 21 × 9 and elliptical kernel with size 21 × 21.

3. Process experimental results and corresponding analyses need to be supplemented.

4. The complexity and processing speed of the algorithm are encouraged to report.

5. In comparative experiments, some higher-level classifiers and improved features should be considered, such as neural networks and improved LBP.

6. Deep learning has demonstrated superior performance in the field of pattern classification. The following related work of image processing including classification, detection, and segmentation must be cited in Section 1, including “Faster mean-shift: GPU-accelerated clustering for cosine embedding-based cell segmentation and tracking.” Medical Image Analysis 71 (2021): 102048. “Compound figure separation of biomedical images with side loss,” Deep Generative Models, and Data Augmentation, Labelling, and Imperfections. Springer, Cham, 2021. 173-183. “Pseudo RGB-D Face Recognition,” in IEEE Sensors Journal, vol. 22, no. 22, pp. 21780-21794, 15 Nov.15, 2022. “Improvement of generalization ability of deep CNN via implicit regularization in two-stage training process,” IEEE Access, vol. 6, pp. 15844-15869, 2018.

Round 2

Reviewer 1 Report

Its better now as compared to first version, just a minor mistake, table 6 is presented after table 7 in paper. Need to fix the table numbering and relevant citing within text. Other than that, its good to go.

Reviewer 2 Report

All the comments have been well revised and thus this paper can be accepted for publication.